# Evaluation of the Corrosion Behavior of Reinforced Concrete with an Inhibitor by Electrochemical Impedance Spectroscopy

**DOI:** 10.3390/ma14195508

**Published:** 2021-09-23

**Authors:** JangHyun Park, MyeongGyu Jung

**Affiliations:** 1Korea Institute of Future Convergence Technology, Hankyong National University, Anseong 17579, Korea; parkjh@hknu.ac.kr; 2Industry Academic Cooperation Foundation, Hankyong National University, Anseong 17579, Korea

**Keywords:** corrosion, rebar, reinforced concrete, electrochemical, inhibitor

## Abstract

In this study, the effect of NaCl and LiNO_2_ content on the deterioration of embedded rebars in concrete due to corrosion was examined by measuring the natural potential and impedance. Wet–dry cycles were performed to accelerate the corrosion of embedded rebars in reinforced concrete, following which the potential and impedance corresponding to the cycles were measured. For the reinforced concrete containing only NaCl, the passive film of the embedded rebar surfaces deteriorated after two weeks of accelerated corrosion, and its polarization resistance decreased. When 0.6 M LiNO_2_ per NaCl was added, the reinforced concrete deteriorated at the same rate as the normal embedded rebars, and the polarization resistance was higher than the initial values. When 1.2 M LiNO_2_ per NaCl was added, the passive film of the embedded rebars remained intact even after 10 weeks of accelerated corrosion, protecting the rebars from deterioration.

## 1. Introduction

The use of reinforced concrete with embedded rebars has been increasing with the rapid advancement of the industry. Corrosion of rebars embedded in reinforced concrete is one of the major factors that affects the durability of concrete structures. In addition, despite the layering of concrete, the rebars embedded within concrete corrode, thereby making it easy for chloride ions to penetrate [1,2].

Embedded rebars are protected from corrosion by stable passive films formed in the strong alkaline (pH > 12.5) environment of concrete [3,4]. However, in recent years, the usage of crushed and washed sea sand mixtures in concrete has led to a scarcity of natural aggregates, such as river gravel and sand [5,6]. Particularly, in terms of sea sand, the presence of chloride ions in the form of NaCl and MgCl_2_ destabilizes the passive films around the embedded rebars, leading to the corrosion of the rebars from the electrochemical potential [7,8,9,10].

The volume of the rebars increases as corrosion propagates and accelerates due to the corrosion products, resulting in concrete cracks. Extensive literature on the corrosion of embedded rebars and its measurements are available [11,12]. Examples of electrochemical techniques adopted to analyze the same include the spontaneous polarization method, linear polarization resistance measurements, and electrochemical impedance spectroscopy (EIS). EIS has been widely used as it allows one to quantitatively calculate the degree of corrosion and continuously observe the crack behavior of rebars [13,14,15,16,17].

To establish fundamental research data, the electrochemical corrosion behavior of embedded rebars in concrete and the effects of chloride ions and corrosion inhibitors are analyzed using electrochemical techniques. Furthermore, to reduce the time required for corrosion of the rebars, a corrosion acceleration method was implemented. Corrosion acceleration was applied to the reinforced concrete specimen, and the characteristic changes in the open circuit potential (OCP) and impedance of the rebar embedded in the concrete, according to the corrosion acceleration time, were observed [18,19]. 

The changes in the impedance, solution resistance (R_s_), and polarization resistance (R_p_) in the embedded rebars of the reinforced concrete were observed over several cycles of accelerated corrosion. The effects of chloride ions and the corrosion inhibitor content on the corrosion behavior of the embedded rebars were subsequently compared and analyzed. The results were used to evaluate the anticorrosive performance of the inhibitor and suggest the appropriate amount of inhibitor used.

## 2. Materials and Methods

### 2.1. Materials

Type 1 ordinary Portland cement (OPC), compliant with the ASTM C 150 standard [20], with a density of 3.15 g/cm^3^ (Ssangyong Company, Seoul, Korea) was used in this study. Table 1 details the chemical composition of OPC.

Domestically sourced crushed stones were used as coarse aggregates, and domestically sourced washed sand was used as the fine aggregate. Polycarbonate admixtures were obtained from S.P (Dongnam company, Pyeongtaek, South Korea). Moreover, NaCl (Duksan company, Ansan, Korea) was used as the source of the chloride ions and LiNO_2_ (Honjo Chemical, Tokyo, Japan) was used as the corrosion inhibitor [21,22].

KS D 3504 D13 rebars (SD400, Hyundai steel, Seoul, Korea) were used for reinforcing concrete [23], and were preprocessed with sandpaper and acetone (Duksan Company, Ansan, Korea) to remove any preexisting corrosion products and accelerate corrosion by peeling off the coating on the surface of the rebar. Furthermore, the surface area not in contact with the concrete was coated with epoxy (Samhwa Paints, Ansan, Korea).

### 2.2. Concrete Mix Proportion

To analyze the effect of the polycarbonate additives on the corrosion behavior of the embedded rebars in concrete, four mixtures of varying concentrations were produced; details of the same are provided in Table 2 and Table 3. The control groups included concrete without introducing chloride ions (normal) and concrete with 1.2 kg/m^3^ chloride ions (C12). Additionally, 0 M LiNO_2_, 0.6 M LiNO_2_ (C12N6), and 1.2 M LiNO_2_ (C12N12) were added to the concrete samples containing chloride ions. Table 2 shows the concrete mix proportions, while Table 3 shows the concentrations of the NaCl and LiNO_2_ added.

### 2.3. Specimens

To measure the corrosion behavior of the rebars embedded in concrete and evaluate the corrosion area, a reinforced concrete specimen was prepared. The reinforced concrete specimens were prepared by fixing a ϕ13 mm rebar (SD-400 deformed) at the center of a Ø100 × 200 mm cylindrical mold and then pouring the concrete into the mold. After cleaning the rebar surface by grit blasting, an epoxy coating was applied to the surface to observe the penetrating effect of the NaCl from the concrete immersed in the NaCl solution; only a length of 100 mm at the center of the concrete specimen was exposed. Figure 1 illustrates the fabricated reinforced concrete specimen. The specimen was demolded 24 h later and cured for 28 days.

### 2.4. Accelerated Corrosion of Reinforced Concrete

As one of the test methods to accelerate corrosion of the reinforced bars, the wetting-drying process was used. Wet–dry cycles accelerate corrosion due to the presence of excess water, and high temperature and humidity, thereby facilitating the diffusion of oxygen. However, as the water content increases, the diffusion of oxygen decreases; accordingly, the temperature and humidity are lowered so as to encourage diffusion and promote further corrosion. This approach was applied in the cycles performed herein to accelerate corrosion of the embedded rebars in concrete [24,25]. Figure 2 illustrates the cycle conditions. 

A temperature and humidity chamber was used to maintain a constant wet period (temperature of 65 °C, relative humidity of 90%) for three days and a constant dry period (15 °C, 60%) for four days; hence, each wet–dry cycle was 1-week-long [26]. At the end of each cycle, the test specimen was immersed in 3.5 wt% NaCl solution for 24 h to stabilize its open circuit potential before measuring its impedance. Accelerated corrosion started immediately following the experiment.

### 2.5. Electrochemical Measurement

According to the electrochemical theory, a simple electrochemical electrode system can be described by an equivalent circuit [14,15,16,17], as shown in Figure 3.

Here, R_s_ denotes the solution resistance, C_dl_ is the capacitance of an electrical double layer, and R_p_ is the polarization resistance by the charge transfer reaction. Figure 4 shows a Bode plot corresponding to the impedance results obtained by using the circuit in Figure 3.

R_s_ corresponds to the impedance at the highest frequency and R_s_ + R_p_ corresponds to the impedance at the lowest frequency. For the rebar-embedded concrete, the R_p_ of the rebar accounts for contributions from the resistance of the film R_f_ and the charge transfer resistance R_ct_ [14,15,16,17]. Therefore, the corrosion resistance of the rebars can be represented by R_p_. In EIS, it can be assumed that the value of the Bode plot at 100 kHz is equal to R_s_ and that at 0.1 Hz is equal to R_s_ + R_p_. Therefore, R_p_ can be calculated.

EIS was performed to measure the impedance of the reinforced concrete after each wet–dry cycle [27,28,29,30] using a potentiostat (PGSTAT302N, Metrohm Autolab B.V., Utrecht, The Netherlands). A three-electrode system was employed, using the rebar as a working electrode (WE), SUS304 as a counter electrode (CE), and Ag/AgCl as a reference electrode (RE). Additionally, before performing EIS for every week of the wet–dry cycle, the rebar was immersed in 3.5 wt% NaCl solution at 25 °C. The experiment is illustrated and summarized in Figure 5 and Table 4 [31]. 

For EIS, the electrode was allowed to stabilize to an open circuit immersed in a 3.5 wt% NaCl solution for 24 h. An AC voltage in the range of 10^−1^ Hz−10^5^ Hz was applied using a potentiostat. The data were subsequently analyzed using Nova software (Metrohm Autolab B.V., Utrecht, The Netherlands).

## 3. Result and Discussion

### 3.1. OCP Results

Metals used in rebars have a unique equilibrium electrode potential. This value is negatively related to the RE and represents the likelihood of the metal to ionize. This phenomenon is known to reflect its degree of corrosion [32,33]. Table 5 shows the criteria for determining rebar corrosion according to the rebar potentials by the type of RE in ASTM C 876 [34] used to determine rebar corrosion.

Accordingly, the equilibrium electrode potential of the rebar in the concrete was measured to quantify the degree of corrosion of the rebar. The OCP of the rebar as a function of the accelerated corrosion time is shown in Figure 6.

For the normal concrete, the OCP did not change significantly as compared to the other samples and remained stable between −0.25 and −0.15 V until the end of 10 weeks of accelerated corrosion, whereby it decreased to −0.315 V. For the C12N6 concrete, the OCP decreased with time from −0.1 to −0.284 V, at a rate much lower than that in the case of the concrete containing only chloride ions. This can be attributed to LiNO_2_, which acts to retard the degradation of the stable passive film that forms around the embedded rebar owing to the strong alkaline nature of the concrete mixture [22,35,36]. For the normal and C12N6 concrete samples, the corrosion status was uncertain until after 9 weeks of cycles, where it entered the initial phase of corrosion, as per ASTM C 876 [34].

For the C12N12 concrete with 1.2 kg/m^3^ chloride ions and 1.2 M LiNO_2_, the OCP remained stable between −0.05 and −0.15 V. The OCP after 10 weeks of cycles was measured to be −0.142 V, corresponding to less than 10% corrosion as per ASTM C 876, implying that no significant corrosion had taken place. This is because the use of an LiNO_2_ concentration equal to or higher than 1.2 M inhibits the degradation of the passive film surrounding the embedded rebar.

### 3.2. Electrochemical Results

The Bode modulus plot shows the impedance Z as a function of the applied frequency. The plots can be categorized by the impedance at a high frequency and the impedance at a low frequency. At a high frequency, the impedance can be used to determine the electrolyte resistance based on the electrode reaction time, and at a low frequency, the polarization resistance of the rebars due to the diffusion and movement of the substances can be identified [27,28,29].

The Bode modulus plots of all test specimens at the selected accelerated corrosion cycles are shown in Figure 7.

As the curves in each Bode modulus plot change according to the accelerated corrosion cycle, the polarization resistance R_p_ can be calculated for the embedded rebars. The calculated R_p_ values are shown in Table 6 and plotted in Figure 8.

For the normal, C12, and C12N6 concrete specimens, the R_p_ of the embedded rebars increased until the second week of the accelerated corrosion cycles and decreased after the fourth week. This can be attributed to the degradation of the passive films of all embedded rebars as they enter the initial stages of corrosion. 

Notably, the R_p_ of the embedded rebar in the C12N12 concrete continued to increase until the fourth week of the accelerated corrosion cycles and decreased after the sixth week. Hence, the addition of 1.2 M LiNO_2_ per added chloride ion enables better protection of the passive film on the embedded rebars as compared to the concrete without chloride ions. Further, 1.2 M LiNO_2_, or more, must be added to the concrete with 1.2 kg/m^3^ chloride ions to retain the protection capability of LiNO_2_.

## 4. Conclusions

In this study, the effects of adding chloride ions and corrosion inhibitors to concrete on the embedded rebars within concrete were analyzed and compared. To accelerate the corrosion of the embedded rebars, wet–dry cycles were performed, and the natural potential and impedance were measured at the end of each 1-week wet–dry cycle. The results are presented as follows: The embedded rebar in the C12 concrete underwent the most rapid deterioration; the rebars embedded in the normal and C12N6 concrete were corrosion initiated at the ninth week of the accelerated corrosion cycle. However, the embedded rebar in the C12N12 concrete was observed to undergo no significant deterioration at the end of 10 weeks of cycles. The passive films were significantly stable with 0.6 M LiNO_2_, although a concentration higher than 1.2 M LiNO_2_ may be required for environments having high salinity in the air, such as coastal structures.The polarization resistance of the normal, C12, and C12N06 concrete increased until the end of the second week of cycles and then decreased, while that of the C12N012 concrete increased until the end of the fourth week and then decreased. This is attributed to the presence of the corrosion inhibitor, LiNO_2_.C12 presented the lowest R_p_ at the end of 10 weeks of cycles at 31.7 Ω cm^3^. The R_p_ values of the normal, C12N06, and C12N12 samples, in increasing order, were 57.6, 113.8, and 223.8 Ω cm^3^, respectively. With 1.2 kg/m^3^ NaCl and more than 0.6 M LiNO_2_ added to concrete, the reinforced concrete is expected to inhibit corrosion more than the normal concrete.The performance of the corrosion inhibitor and the corrosion behavior of the embedded rebars in concrete were analyzed using wet–dry cycles and electrochemical techniques. In addition, it was confirmed that the corrosion of the rebar embedded in the concrete can be protected from the penetration of chloride ions using an appropriate inhibitor. Through electrochemistry and microanalysis of in/organic inhibitors, their mechanisms toward inhibiting corrosion may be considered in future research.

## Figures and Tables

**Figure 1 materials-14-05508-f001:**
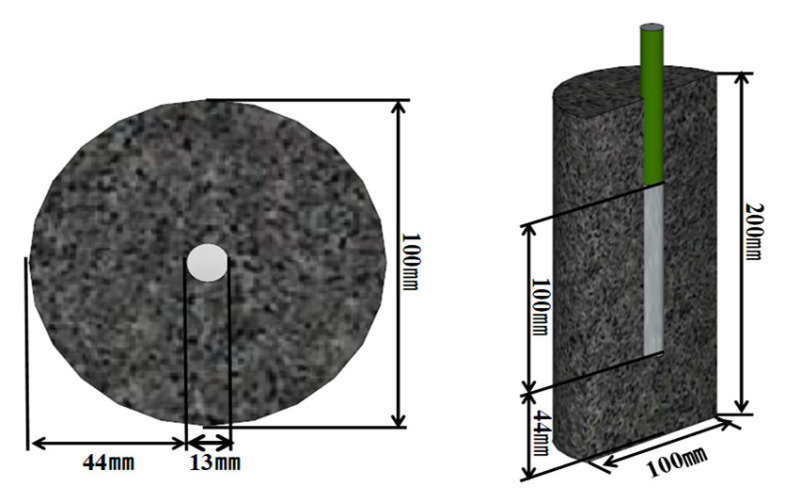
Schematic of specimen fabricated for electrochemical measurement.

**Figure 2 materials-14-05508-f002:**
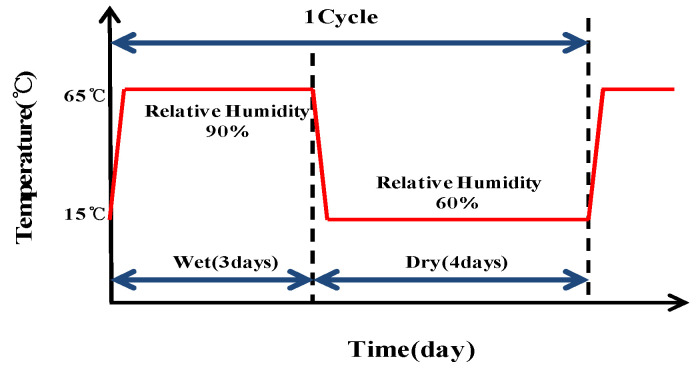
Corrosion acceleration cycle (wet–dry method).

**Figure 3 materials-14-05508-f003:**
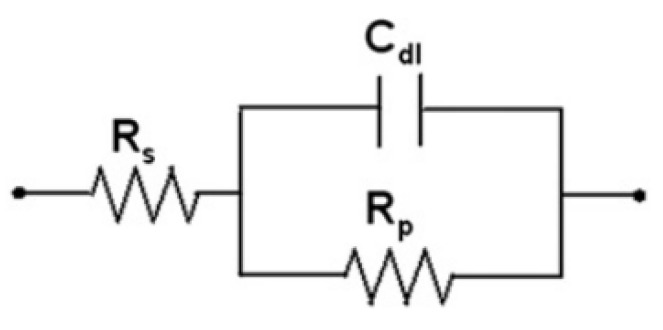
Schematic of a simple electrochemical electrode system.

**Figure 4 materials-14-05508-f004:**
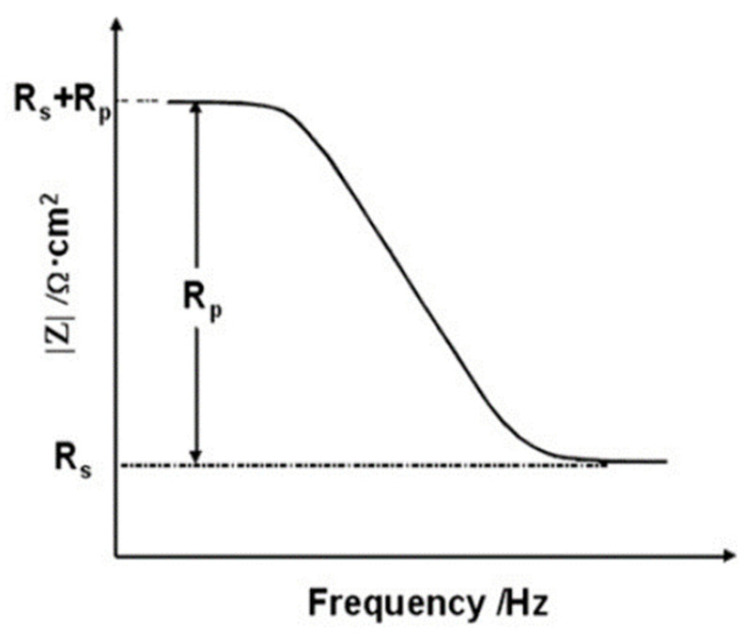
Bode plot for electrochemical impedance spectroscopy measurement.

**Figure 5 materials-14-05508-f005:**
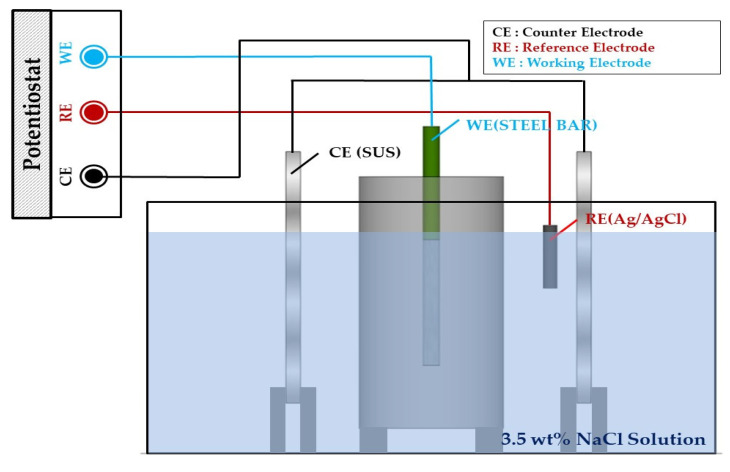
Experimental setup for the EIS experiment.

**Figure 6 materials-14-05508-f006:**
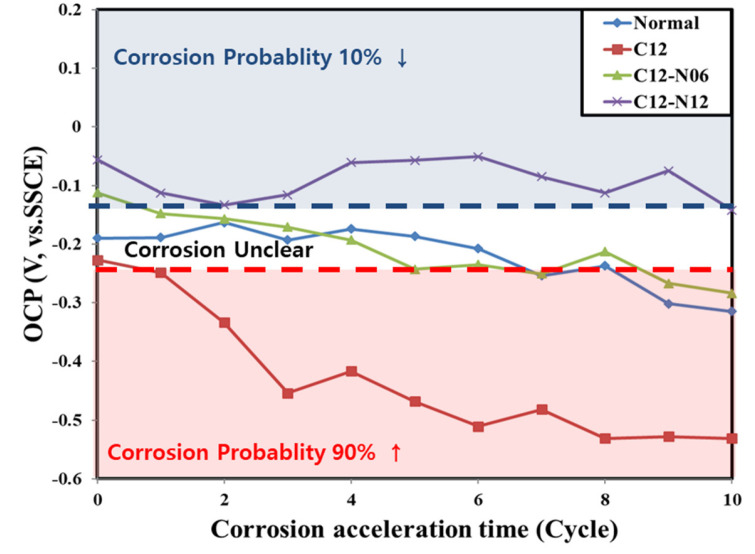
OCP of the rebars embedded in concrete according to the corrosion acceleration cycle.

**Figure 7 materials-14-05508-f007:**
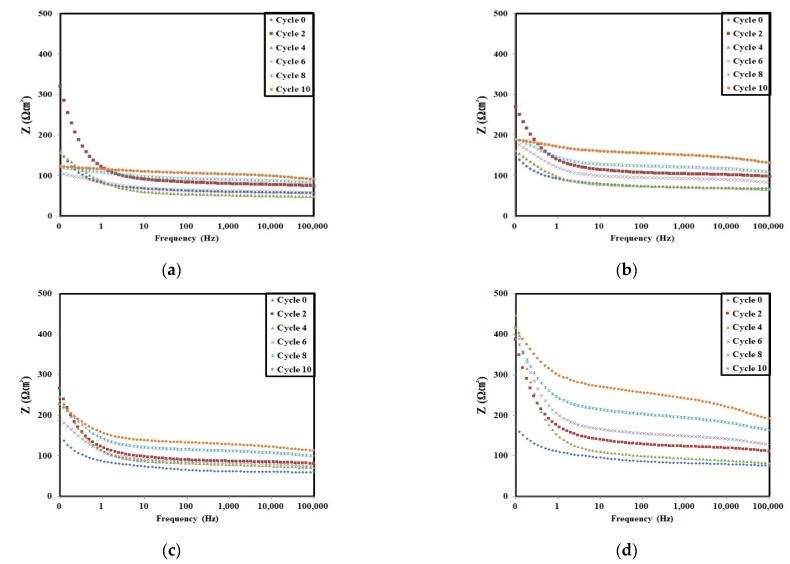
Bode modulus plot of the rebars embedded in concrete according to the corrosion acceleration cycle: (**a**) normal, (**b**) C12, (**c**) C12N6, and (**d**) C12N12.

**Figure 8 materials-14-05508-f008:**
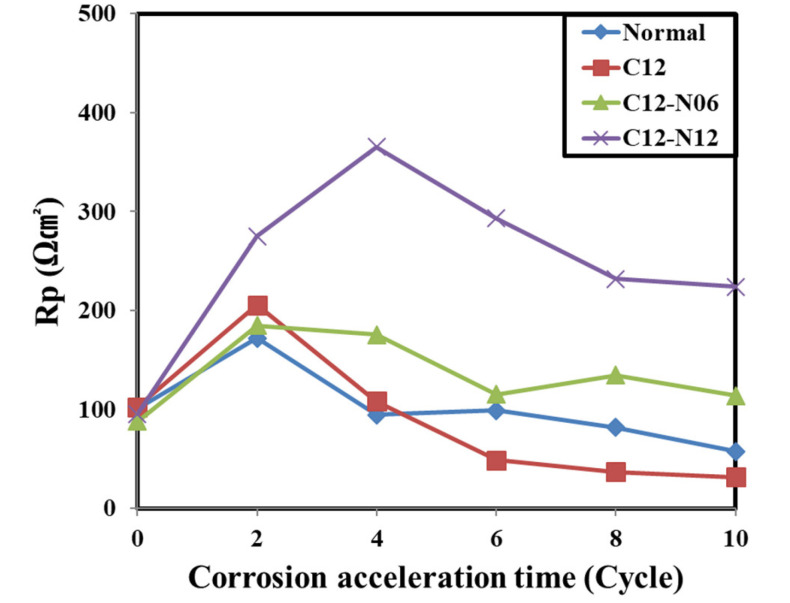
Variation in R_p_ with accelerated corrosion cycle.

**Table 1 materials-14-05508-t001:** Chemical composition of cement.

Name	Chemical Composition (%)									* L.O.I.
	SiO_2_	Al_2_O_3_	TiO_2_	Fe_2_O_3_	CaO	MgO	SO_3_	K_2_O	etc.	
OPC	19.74	5.33	0.30	2.93	61.74	3.78	2.47	0.89	2.82	2.3

* L.O.I.: loss on ignition.

**Table 2 materials-14-05508-t002:** Mix proportion of concrete.

W/C	Unit Weight (kg/m^3^)				
	Water	Cement	Gravel	Sand	* S.P.
50%	172	344	941	711	1.03

* S.P.: superplasticizer admixture.

**Table 3 materials-14-05508-t003:** Amounts of NaCl and LiNO_2_ added to concrete.

No.	Name	NaCl (kg/m^3^)	LiNO_2_	
			Molar Ratio (NO_2_^−^/Cl^−^)	Addition (kg/m^3^)
1	Normal	0.0	0.0	0.0
2	C12	1.2	0.0	0.0
3	C12N6		0.6	4.3
4	C12N12		1.2	8.6

**Table 4 materials-14-05508-t004:** Electrochemical experiment conditions.

Frequency range	10^5^–10^−1^ Hz
Specimen size	Ø100 × 200 mm^2^
Cover concrete	44 mm
WE	Ø13 mm rebar (SD 400)
RE	Ag/AgCl
CE	STS 304

**Table 5 materials-14-05508-t005:** ASTM C 876 criteria for corrosion.

Potential of Rebar (mV)			Corrosion Probablity
CSE	SSCE	SHE	
<−500	<−426	<−184	Severe
<−350	<−276	<−34	90% ↑
−350~−200	−276~−126	−34~+116	50% ↓
>−200	>−126	>+116	10% ↓

**Table 6 materials-14-05508-t006:** Variation in R_p_ with accelerated corrosion cycle.

Name	R_p_ with Accelerated Corrosion Cycle (Ω cm^2^)					
	0	2	4	6	8	10
Normal	100	172	94	99	82	58
C12	103	206	108	49	37	32
C12-N06	88	185	176	115	135	114
C12-N12	96	275	366	293	232	224

## Data Availability

The data presented in this study are available on request from the corresponding author.

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
