# Peer review of "Evaluation of the Corrosion Behavior of Reinforced Concrete with an Inhibitor by Electrochemical Impedance Spectroscopy"

_materials, 2021, doi:10.3390/ma14195508_

Round 1

Reviewer 1 Report

Comments to the Authors

I have reviewed the manuscript “Electrochemical Impedance Spectroscopy Evaluation of the Corrosion Behavior of Reinforced Concrete with an Inhibitor” by JangHyun Park and MyeongGyu Jung. The work is well-organised and needs some revisions to be published in Materials. Here are my comments.

  1. Revise some typos in the text.

  1. Within the Introduction and text, add some references dealing with degradation and durability of RC structures such as:

Castaldo, P., Palazzo, B., & Mariniello, A. (2017). Effects of the axial force eccentricity on the time-variant structural reliability of aging rc cross-sections subjected to chloride-induced corrosion. Engineering Structures, 130, 261-274.

Biondini, M. Vergani Deteriorating beam finite element for nonlinear analysis of concrete structures under corrosion Struct Infrastruct Eng, 11 (4) (2015), pp. 519-532

  1. Add some comments on the calculation of the corrosion probabilities reported in Figure 6.

  1. Add some comments regarding the design possibilities to improve the safety and sustainability of structures.

Author Response

Response to Reviewer 1 Comments

Point 1. Revise some typos in the text.

 Response 1: Thank you for your review. The responses to your comments are provided below.

The entire paper was revised by a native speaker in concrete technology. Moreover, overall sentence and words have been corrected with a focus on the scientific point.

Several terms, phrases, and sentences were revised in the manuscript, and it was not possible to mention every single one of them in this response letter. Kindly review the revised manuscript.

Point 2. Within the Introduction and text, add some references dealing with degradation and durability of RC structures such as:

Castaldo, P., Palazzo, B., & Mariniello, A. (2017). Effects of the axial force eccentricity on the time-variant structural reliability of aging rc cross-sections subjected to chloride-induced corrosion. Engineering Structures, 130, 261-274.

Biondini, M. Vergani Deteriorating beam finite element for nonlinear analysis of concrete structures under corrosion Struct Infrastruct Eng, 11 (4) (2015), pp. 519-532

Response 2: The References and Introduction have been revised by adding two references recommended by the reviewer.

  1. Castaldo, P.; Palazzo, B.; MArinello, A. Effects of the axial force eccentricity on the time-variant structural reliability of ag-ing r.c. cross-sections subjected to chloride-induced corrosion, Engineering Structure, 2017, 130, 261–274
  2. Biondini, F.; Vergani, M. Deteriorating beam finite element for nonlinear analysis of concrete structures under corrosion, Deteriorating beam finite element for nonlinear analysis of concrete structures under corrosion, Structure and Infrastructure Engineering, 2015, 11(4), 519–532

Point 3. Add some comments on the calculation of the corrosion probabilities reported in Figure 6.

Response 3: Note that the corrosion probability calculations have been added to Lines 146–151.

“Table 5 shows the criteria for determining rebar corrosion according to the rebar potentials by the type of reference electrode in ASTM C 876 used to determine rebar corrosion.”

Table 5. ASTM C 876 criteria for corrosion.

Potential of Rebar (mV)

Corrosion

Probablity

CSE

SCE

SHE

<  −500

<  −426

<  −184

Severe

<  −350

<  −276

<  −34

90% ↑

−350 ~ −200

−276 ~ −126

−34 ~ +116

50% ↓

>  −200

>  −126

>  +116

10% ↓

Point 4. Add some comments regarding the design possibilities to improve the safety and sustainability of structures.

Response 4: Conclusion 4 has been corrected to reflect the comments.

Corrected on Line 233–235:

“it was confirmed that the corrosion of the rebar embedded in the concrete can be protected from the penetration of chloride ions using an appropriate inhibitor.”

Reviewer 2 Report

Dear authors, I appreciate positively the effort to carry out this research and the results you obtained. However, I think it would be useful to supplement with data on corrosion inhibitors commonly used in the construction industry (in the Introduction). You should also highlight the usefulness of this research: statistics on losses caused by corrosion of reinforcing elements of concrete structures, where this type of concrete added with corrosion inhibitors can find its place, etc.

I am attaching your article in which I have made a number of other observations, mainly related to the translation.

Author Response

Response to Reviewer 2 Comments

Thank you for your review. The responses to your comments are provided below.

The entire paper was revised by a native speaker in concrete technology. Moreover, overall sentence and words have been corrected with a focus on the scientific point.

Several terms, phrases, and sentences were revised in the manuscript, and it was not possible to mention every single one of them in this response letter. Kindly review the revised manuscript.

Point 1 : Please translate better, because the sentence is not clear.

Response 1: The passage has been revised for clarity, as shown in lines 29–31.

“However, in recent years, the usage of crushed and washed sea sand mixtures in concrete has led to a scarcity of natural aggregates, such as river gravel and sand [5,6].”

Point 2 : please explain the abbreviation

Point 3 : please give more informations about polycarbonate admixtures

Response 4: Table 2 shows the unit weight of the materials used for concrete mixing.

W: water, C: Cement, G: Gravel (coarse aggregate), S: Sand (fine aggregate), S.P: Spuer-plasticizer (water reducer)

Point 5 : please explain why you chose the length of 100 mm inside the specimen to be coated with epoxy

Response 7: This passage has been revised accordingly on Lines 167–168.

“at a rate much lower than that in the case of the concrete containing only chloride ions.”

Point 8 : please translate better

Response 8: This has been corrected in Lines 211.

“concrete were analyzed and compared.”

Response 9: This has been corrected in Lines 215–217.

“the rebars embedded in the normal and C12N6 concrete were corrosion initiated at the ninth week of the accelerated corrosion cycle.”

Point 10 : why? - plese explain better

Response 10: This has been corrected in Lines 219–221.

“although a concentration higher than 1.2 M LiNO2 may be required for environments having high salinity in the air, such as coastal structures.”

Point 11 : it is about the corrosion inhibitor, I believe, not about the reinforcement - please make the corrections

“With 1.2 kg/m3 NaCl and more than 0.6 M LiNO2 added to concrete, the reinforced concrete is expected to inhibit corrosion more than the normal concrete.”

Point 12 : It would be interesting in another research to complete this type of analysis with imaging (SEM or other type) through which to complete the results with data about the thickness of the corroded layer.

Reviewer 3 Report

The manuscript presents useful information, and only requires relatively minor revision prior to publication.  My detailed comments and suggestions are:
1) Lines 23-24:  The meaning of withstand deterioration is unclear.  What I think you wish to say something like “Concrete is normally reinforced with steel to fist its low tensile strength.”  Similarly, the meaning of layering is unclear.

2) Introduction:  The discussion should put greater emphasis on the justification underpinning use of accelerated corrosion’s tests.  To remember is that not only people who are experts in the topic will read your paper.

3) Table 1:  Make percentages add up to exactly 100%.

4) Lines 63-67:  Given you cleaned regards, you need to justify how/why your results will be applicable to normal practice on construction sites.

5) Sections 2.3 and 2.4:  Explain the logic underpinning what you did, rather than simply saying what you did.

6) Table 5:  Quoting results to four figure precision implies an unrealistic degree of experimental accuracy.  Quoting values to three figure precision would be more realistic.

7) 

Author Response

Response to Reviewer 3 Comments

Thank you for your review. The responses to your comments are provided below.

The entire paper was revised by a native speaker in concrete technology. Moreover, overall sentence and words have been corrected with a focus on the scientific point.

Several terms, phrases, and sentences were revised in the manuscript, and it was not possible to mention every single one of them in this response letter. Kindly review the revised manuscript.

Point 1 :  Lines 23-24:  The meaning of withstand deterioration is unclear.  What I think you wish to say something like “Concrete is normally reinforced with steel to fist its low tensile strength.”  Similarly, the meaning of layering is unclear.

Response 1: This has been corrected in Lines 24–27.

“Corrosion of rebars embedded in reinforced concrete is one of the major factors that affects the durability of concrete structures. In addition, despite the layering of concrete, the rebars embedded within concrete corrode, thereby making it easy for chloride ions to penetrate [1,2].”

Point 2 :  Introduction:  The discussion should put greater emphasis on the justification underpinning use of accelerated corrosion’s tests. To remember is that not only people who are experts in the topic will read your paper.

Response 2: The reason for evaluating the corrosion behavior of the rebar embedded in concrete using corrosion acceleration and the purpose of this study were corrected and added to the revised manuscript.

“Furthermore, to reduce the time required for corrosion of the rebars, a corrosion acceleration method was implemented. Corrosion acceleration was applied to the reinforced concrete specimen, and the characteristic changes in the open circuit potential (OCP) and impedance of the rebar embedded in the concrete, according to the corrosion acceleration time, were observed [18,19].”

“The results were used to evaluate the anticorrosive performance of the inhibitor and suggest the appropriate amount of inhibitor used.”

The text has been corrected in Lines 44–48 and 53–54.

Point 3 : Table 1:  Make percentages add up to exactly 100%.

Response 3: Note that L.O.I has not been included in the chemical composition.

The text has been corrected in Table 1.

Point 4 : Lines 63-67:  Given you cleaned regards, you need to justify how/why your results will be applicable to normal practice on construction sites.

Response 4:

There are two main reasons to clean the rebar surface:

1) To reduce the error by removing the corrosion product on the surface of the rebar

2) To accelerate corrosion by peeling off the coating on the surface

The relevant text has been corrected in Lines 69–70.

Point 5 : Sections 2.3 and 2.4:  Explain the logic underpinning what you did, rather than simply saying what you did.

Response 5: The text has been revised to include the purpose and rationale for 2.3 and 2.4.

This has been corrected in Lines 87–88 (2.3)

This has been corrected in Lines 99–100.

Point 6 : Table 5:  Quoting results to four figure precision implies an unrealistic degree of experimental accuracy.  Quoting values to three figure precision would be more realistic.

Response 6: The precision of the data has been corrected to an integral range.

You can find the corrections in Table 6.

Round 2

Reviewer 1 Report

Comments to the Authors

I have reviewed the revised manuscript “Electrochemical Impedance Spectroscopy Evaluation of the Corrosion Behavior of Reinforced Concrete with an Inhibitor” by JangHyun Park and MyeongGyu Jung. The revised text can be published in Materials.